# The Benefits of Including Hummus and Hummus Ingredients into the American Diet to Promote Diet Quality and Health: A Comprehensive Review

**DOI:** 10.3390/nu12123678

**Published:** 2020-11-28

**Authors:** Evan J. Reister, Lynn N. Belote, Heather J. Leidy

**Affiliations:** 1Department of Nutrition Science, Purdue University, West Lafayette, IN 47907, USA; ereister@purdue.edu; 2Sabra Dipping Company, LLC, 777 Westchester Ave., 3rd Floor, White Plains, NY 10604, USA; lbelote@sabra.com; 3Department of Nutritional Sciences, University of Texas at Austin, Austin, TX 78723, USA; 4Department of Pediatrics, University of Texas at Austin, Austin, TX 78723, USA

**Keywords:** hummus, diet quality, type 2 diabetes, cardiovascular disease, obesity, health

## Abstract

Over the last decade, hummus has become an increasingly popular food. Given the ingredients (i.e., primarily chickpeas and tahini), nutrient composition, versatility, and acceptability, hummus can play a unique role when included in the American diet, to promote diet quality and improve health. The purpose of this paper is to provide a comprehensive review of the scientific evidence examining the effects of acute and long-term consumption of hummus and hummus ingredients on diet quality and risk factors related to type 2 diabetes, cardiovascular disease, and obesity. In addition, food pattern/menu modeling is included to illustrate the potential nutritional impact of consuming hummus to meet dietary guidelines. In general, the consumption of hummus and/or its respective ingredients has been shown to improve postprandial glycemic control, fasting lipids, appetite control, and daily food intake compared to other commonly consumed foods. The incorporation of hummus into the American diet can also improve diet quality by replacing foods higher in saturated fats, sodium, or added sugars. Collectively, these findings support the addition of hummus and/or hummus ingredients as an important component of a healthy dietary pattern.

## 1. Introduction

Two decades ago, most Americans had never heard of hummus, much less tasted it, or included it as part of a healthful diet. Now, hummus can be found in the refrigerators of approximately 25% of American homes and is increasingly available on restaurant menus [1]. A number of potential factors have contributed to the surge in hummus popularity in the United States. There is a continued desire to purchase and consume foods that: (1) taste good; (2) promote health and well-being; (3) help meet current plant-based dietary recommendations; and (4) are easy to include with other healthy foods [2]. Further, hummus fits into a variety of healthy eating patterns, including the Mediterranean dietary pattern [3], and is used as part of the protein foods and vegetable categories within the United States Department of Agriculture (USDA) MyPlate [4]. Given the versatility, palatability, and potential health benefits, hummus can play a unique role when included in the American diet, to promote health and diet quality.

Americans continue to consume less vegetables than the current dietary guideline recommendations. In 2015, only one in 10 Americans met the 2015–2020 Dietary Guidelines for Americans’ vegetable recommendations of 2.5 cups/day within a 2000 calorie diet [4,5]. Within these guidelines, it is recommended that at least 1.5 cups/wk of vegetables should be from legumes; however, mean intake of legumes is less than half of the recommended amount in American adults [4]. Although one serving (i.e., two tablespoons) of hummus per day could contribute to meeting the weekly recommendations, it is unclear whether there is sufficient evidence supporting the specific effects of hummus on health outcomes.

Thus, the purpose of this paper is to provide a comprehensive review of the scientific evidence concerning the inclusion of hummus (and the respective chickpea and tahini ingredients) into the American diet to promote diet quality and health. Specific health outcomes of interest include type 2 diabetes, cardiovascular disease, and obesity risk factors.

## 2. Methodology for the Comprehensive Review

The search terms for the independent variables included “hummus”, “chickpeas”, “sesame”, “tahini”, and “Mediterranean diets.” The search terms for the dependent variables related to diet quality and type 2 diabetes, cardiovascular disease, and obesity risk factors included “diet quality”, “food intake”, “vegetable consumption”, “health”, “glycemic response”, “blood glucose”, “diabetes”, “cardiovascular disease”, “blood lipids”, “LDL Cholesterol”, “HDL Cholesterol”, “triglycerides”, “inflammation”, “blood pressure”, “weight gain”, “body fat”, “BMI”, and prevalence of obesity.

Searches of electronic databases were carried out between August 2019 and November 2019, and included PubMed and Scopus. In addition, references from existing reviews and select articles were examined to supplement the electronic search. 

This review was limited to articles published in English in peer-reviewed journals, and included the following criteria: (1) all age groups; (2) all diseases/conditions; (3) any study that included hummus or one (or more) ingredients; and (4) studies that included diet quality, type 2 diabetes, cardiovascular disease, and obesity risk factor outcomes listed above. All data are presented as mean ± SEM.

## 3. Nutrient Intake and Diet Quality

While there is currently no standard definition for hummus, the two primary ingredients include chickpeas and tahini, both of which contain essential nutrients and bioactive microconstituents. A complete nutrient profile for hummus as well as chickpeas and tahini is found in Table 1. Chickpeas are pulses, which are the edible seeds of plants in the legume family. Chickpeas contain protein, fiber, resistant starch, and unsaturated fatty acids along with a variety of vitamins and minerals, including riboflavin, niacin, thiamin, folate, potassium, phosphorus, and calcium [6,7]. Tahini, a paste made from toasted ground sesame, is composed of unsaturated fatty acids, antioxidant lignans and tocopherols, and key minerals including calcium and phosphorus [8]. Since hummus is a combination of chickpeas and tahini, it could help achieve key nutrient and vegetable or protein food group recommendations within USDA’s MyPlate [9,10].

In addition to improving nutrient intake, a few studies support the consumption of hummus or hummus ingredients to improve diet quality through the replacement of energy-dense, nutrient-poor foods. In a recently published randomized crossover study by Reister et al. [11], 39 healthy adults were provided with 240 kcal afternoon snacks containing 2 servings (57 g) of hummus and a serving of pretzels, an isocaloric higher-sugar (granola bar) snack, or no snack for 7 days/snack. The afternoon hummus snack led to a 250 kcal reduction in subsequent evening snacking on high-sugar dessert foods compared to the higher-sugar snack or no snacking. In two other crossover studies, the inclusion of chickpeas (104 g/day) into the diet was examined over 12 weeks followed by 4 weeks of habitual diet without chickpeas in adults. In both studies, the daily consumption of chickpeas led to voluntary reductions in all food groups compared to the habitual diet. Of particular interest is the reduction in nutrient-poor, energy-dense snacks [12,13]. Thus, these data illustrate improvements in diet quality as a result of consuming hummus or hummus ingredients. Potential mechanisms for these observations have been examined and are discussed in the Weight Management Section of this review.

## 4. Type 2 Diabetes Risk Factors

Over 100 million Americans have diabetes or prediabetes and exhibit increased risks of developing metabolic syndrome; insulin resistance; non-alcoholic fatty liver disease; hypertension; and cardiovascular disease [14,15,16,17,18]. Given that type 2 diabetes is, in most cases, a preventable disease, there is a strong emphasis on identifying healthy dietary patterns and foods/food groups that may improve glucose control to help prevent and/or treat this condition.

Most of the literature assessing the effects of specific foods on glucose control is from acute, single-meal studies in which 2 to 4-h postprandial glucose and insulin concentrations are measured. For example, in a randomized crossover study by Augustin et al. [19], 10 healthy adults consumed varying quantities of hummus (28, 56, 112, 259 g) with and without white bread, compared to consuming white bread alone. Postprandial glucose and insulin responses were assessed during a single 2-h testing day for each meal. Postprandial glucose area under the curve (AUC) was lower following consumption of hummus alone compared to white bread, even when the hummus was carbohydrate-matched (all, *p* < 0.05). The 28 and 112 g servings of hummus also resulted in lower insulin AUC than white bread (both, *p* < 0.05), but the 259 g serving of hummus was not significantly different from the carbohydrate-matched white bread in insulin AUC. However, hummus is hardly ever eaten in isolation, and this study found that the addition of hummus to white bread did not reduce the glycemic response compared to white bread alone. In a more recent study, the glycemic response to an afternoon hummus snack (i.e., hummus and pretzels) was compared to an isocaloric, higher-sugar snack (i.e., granola bars) using 24-h continuous glucose monitoring [11]. The hummus snack led to lower 3-h post-snack blood glucose AUC compared to the higher-sugar snack (both, *p* < 0.05). This indicates that hummus has potential to improve short-term glycemic response, but this is dependent on the food it is paired with and its comparator. 

In examining the various components of hummus, the consumption of chickpeas exhibits several beneficial effects on glucose control. In a randomized crossover study of 15 healthy males by Mollard et al. [20], the consumption of 223 g (300 kcal) of chickpeas resulted in a 47% reduction in 2-h blood glucose AUC compared to isocaloric white bread (*p* < 0.05). Similarly, in a randomized crossover study of 12 healthy females, the consumption of 200 g (218 kcal) of chickpeas resulted in a 36% reduction in 2-h blood glucose AUC compared to isocaloric white bread (*p* < 0.05) [21]. Lastly, a study by Wong et al. [22] involving 15 healthy males found that the consumption of 341 g (300 kcal) of chickpeas resulted in a 36% reduction in 2-h blood glucose AUC compared to isocaloric white bread (*p* < 0.05). The Wong et al. study [22], along with the Mollard et al. study [20], examined blood glucose following consumption of other pulses as well, but found no differences in the 2-h blood glucose AUC between any legumes. However, in a study of 11 healthy adults, the consumption of 100 g of chickpeas was shown to reduce the 1-h mean glycemic response compared to carbohydrate-matched consumption of white bread, black beans, pigeon peas, and mung beans (all, *p* < 0.01) [23]. 

Concerning the effects of chickpea consumption on postprandial insulin responses, the consumption of 200 g of chickpeas in 19 healthy adults led to reductions in 2-h postprandial insulin concentrations, improvements in insulin resistance, and improvements in β-cell function as evidenced by the reductions in homeostasis model assessment (HOMA) (both, *p* < 0.05) compared to carbohydrate-matched white or wheat bread [24]. However, when chickpea consumption was examined over the longer-term (i.e., 6 wks) compared to a single day, improvements in insulin sensitivity were not observed [24]. Potential reasons for this discrepancy may be due to the inclusion of chickpea flour within the longer-term study. The processing of the flour may have lessened the low glycemic-index (GI) benefits of the original chickpeas. Thus, more research is needed to explore the effects of chickpeas, particularly chickpea processing, on long-term glycemic control.

Less is known with respect to the effects of tahini consumption on glucose control. In a randomized clinical trial involving 41 patients with type 2 diabetes, the incorporation of 2 tbsp (~28 g) of tahini into a breakfast meal for 6 weeks resulted in slight but non-significant decreases in blood glucose, insulin, and HOMA compared to a habitual diet group [25]. Although these findings suggest that tahini alone has little impact on glucose control, this was a single, acute study. Thus, more research is needed to assess possible beneficial effects of tahini on glucose control.

Hummus and its respective ingredients are considered low glycemic-index (GI) foods. On the glucose reference scale that compares foods to white bread (GI of 100), hummus and chickpeas have GIs ranging from 14 to 28 [19,26]. The low GI characteristics are a result of their low sugar: high fiber/protein/fat content, slow digestibility, and slow rates of absorption [19,21,26]. Specifically, chickpeas, and thus hummus, contain a high amount of the resistant starch amylose [21,24]. Amylose, which contributes between 20 to 45% of the starch content of chickpeas [27], is digested and absorbed more slowly than other food components, thus reducing postprandial blood glucose concentrations. Additionally, the fermentation of amylose is thought to increase gut hormones such as GLP-1 and PYY [28], which in turn improves glucose tolerance and enhances insulin secretion [29]. Additionally, the protein content and amino acid profile of hummus also contributes to its glucose regulatory effects [30,31]. Hummus contains soluble fibers that decrease gastric emptying and increase gastric distention [32], resulting in a decrease in the digestive process and reduced glucose release. Lastly, hummus and its ingredients have high monounsaturated fatty acid (MUFA) and polyunsaturated fatty acid (PUFA) content [19,26,33] which also improve glucose metabolism [34,35,36]. Additionally, tahini is rich in several antioxidant lignans, including sesamin. Sesamin supplementation is found to improve glucose control in humans; although the underlying mechanism is unclear [37] but thought to be related to its antioxidant activity [38]. Overall, in comparison to commonly consumed foods such as white bread, hummus and chickpeas appear to improve glucose control over the short-term, although more long-term studies are needed.

## 5. Cardiovascular Disease Risk Factors

Cardiovascular disease (CVD) is the leading cause of death in America, with over 840,000 deaths attributed to CVD in 2016 [39]. Hypercholesterolemia (high blood cholesterol), hyperlipidemia (high blood lipids), and hypertension (high blood pressure) are three modifiable risk factors that contribute to development of CVD [40,41,42]. Modest reductions in blood lipids (including cholesterol) and blood pressure reduce the relative risk of CVD events [43,44,45]. Diets rich in fruits, vegetables, legumes, fat free and low-fat dairy products, and complex carbohydrates and limited in sugars, salt, and saturated fats are shown to be effective in lowering blood lipids and blood pressure [43,46,47].

Although no studies to date have examined the effects of hummus on blood lipids or blood pressure, several clinical trials demonstrate effects of hummus ingredients on these outcomes. Compared to a habitual diet, the incorporation of chickpeas (104 g/day) into the diet of 45 mildly hypercholesterolemic adults for 12 weeks lowered total cholesterol an average of 0.20 mmol/L (*p* = 0.002) and low-density lipoprotein (LDL) cholesterol an average of 0.19 mmol/L (*p* = 0.01) [12]. In two randomized crossover studies in healthy adults, a 5-week chickpea-supplemented diet (140 g/day) lowered total cholesterol 0.22 mmol/L (*p* = 0.001) [48] and 0.25 mmol/L (*p* < 0.01) [49] compared to a 5-week wheat-supplemented diet. In these studies, LDL cholesterol following chickpea supplementation was decreased by 0.18 mmol/L (*p* = 0.002) [48] and 0.20 mmol/L (*p* = 0.02) [49] compared to the wheat-supplemented diet. Additionally, Mathur et al. [50] examined the effects of substituting chickpeas for wheat-flour or other cereals in a high-fat diet for 55 weeks in 20 healthy males. The incorporation of chickpeas was the only alteration to the lifestyle or dietary pattern, and total calories were the same both before and after chickpea substitution into the diet. The inclusion of chickpeas lowered fasting total cholesterol from 206.4 ± 20.0 mg/100 mL to 160.0 ± 24.1 mg/100 mL (*p* < 0.05), a reduction of 22.5% [50]. Ester cholesterol and beta-lipoprotein were also found to decrease significantly (*p* < 0.001). High-density lipoprotein (HDL) and LDL were not measured in the study. Given that a reduction in total cholesterol of 10% is associated with decreased mortality from CVD in older adults [43], these findings suggest that chickpeas might be a helpful dietary strategy to assist in reducing CVD risks.

The beneficial actions of chickpeas on cholesterol and lipid levels are thought to be due in part to their fatty acid profile. Chickpeas are relatively high in PUFA and low in saturated fat, and the consumption of chickpeas is associated with an increased intake of PUFA (as a percentage of total fat) and a reduced intake of saturated fat (as a percentage of total fat) [10]. Diets high in PUFA and low in saturated fat are effective in decreasing total and LDL cholesterol [51,52,53,54], which is possibly due to an increase in LDL receptors brought about by the diet [55,56].

Besides their fatty acid profile, chickpeas contain a variety of health-promoting phytonutrients, including isoflavones, saponins, and fiber. Isoflavones and other phenolic phytonutrients have strong antioxidant capabilities to scavenge free radicals and may positively influence cholesterol and lipid levels [57,58]. The saponin content in chickpeas is higher than in many other pulses, and this non-phenol phytonutrient is known to form an insoluble complex with cholesterol and thereby inhibit both cholesterol absorption and bile acid reabsorption [57,59]. High levels of phytonutrients or soluble fiber are shown to increase bile acid binding capability and bile acid fecal excretion [60]. Chickpeas appear particularly effective in binding and excreting cholesterol as bile acid [50,61,62]. In addition to the nutritional factors already listed, vegetable protein, oligosaccharides, flavanols, and phytosterols are also thought to contribute to the hypocholesteremic and hypolipidemic effect of chickpeas [63]. All of these nutrients and bioactive compounds work synergistically to modify blood cholesterol and lipid levels.

With respect to tahini, a 6-week randomized control trial was completed in 36 adults with diabetes. The participants were divided into two groups: those that consumed a habitual breakfast (i.e., control group) and those who replaced part of their habitual breakfast with 2 tbsp (~28 g) of tahini [64]. Although the tahini breakfast did not improve total or LDL cholesterol, triglycerides were reduced by an average of −15.3 mg/dL (~10% decrease) in this group compared to the control (*p* = 0.04). HDL cholesterol in the tahini group tended to increase compared to control (*p* = 0.07), increasing by +1.8 mg/dL. In addition, the atherogenic index of plasma (AIP), which is directly related to the risk of atherosclerosis [64], was reduced by 39% following the tahini breakfast, compared to the control (*p* = 0.05). Collectively, these data suggest that tahini could play a role in improving lipid profiles and decreasing CVD risk. 

Tahini is composed of about 40% MUFA and 40% PUFA, and evidence shows that diets enriched with a mixture of MUFA and PUFA are effective in lowering total and LDL cholesterol [52,54,65]. Tahini is also a good source of the antioxidant lignans sesamin and sesamol, both of which have a part in reducing cholesterol. Sesamin inhibits cholesterol absorption and synthesis [66], whereas sesamol improves macrophage cholesterol efflux [67]. Lastly, tahini is rich in other nutritional factors that influence cholesterol, including phytosterols, flavonoids, and fiber. These phytoconstituents are thought to increase cholesterol excretion and hepatic bile acid production as well as improve hepatic antioxidant status [68]. A single serving of tahini likely does not contribute enough phytoconstituents to produce a cholesterol-lowering effect on its own, but in the context of a healthy diet it has potential to serve a cardio-protective role.

## 6. Weight Management

The obesity epidemic continues to be a major public health concern with overweight and obesity rates currently at approximately 70% in 2015–2016 [69]. Obesity, which is intimately related to poor eating and physical activity patterns, increases the risk of developing a number of disease conditions, including type 2 diabetes, cardiovascular disease, and certain cancers [70]. Therefore, weight management strategies are vitally needed to improve overall health. 

No studies to date exist testing whether the incorporation of hummus, chickpeas, or tahini directly influence weight changes. However, one observational study exists that examines the relationship between weight and chickpea and hummus consumption [10]. On average, chickpea and hummus consumers weigh less than non-consumers, with consumers weighing an average of 75.8 ± 1.6 kg compared to 81.9 ± 0.3 kg for non-consumers (*p* = 0.0002). Additionally, body mass index (BMI) and waist circumference were lower for consumers compared to non-consumers. Consumers had an average BMI of 26.4 ± 0.5 kg/m^2^ and a waist circumference of 92.2 ± 1.3 cm, while non-consumers had a BMI of 28.6 ± 0.1 kg/m^2^ and a waist circumference of 97.9 ± 0.3 cm (both, *p* < 0.0002). Despite the limited evidence on weight management, a number of studies have examined whether the consumption of these foods improves indices of weight management.

In the previously mentioned snack study, appetite and satiety were measured before and every 30 min after consumption of an afternoon hummus snack, isocaloric higher-sugar snack, or no snack throughout a 3-h period. The hummus snack led to 70% reductions in hunger, desire to eat, and prospective food consumption compared to no snacking (all, *p* < 0.05), whereas a higher-sugar snack did not [11]. Although the hummus snack increased post-snack fullness compared to no snack, no differences were observed between the hummus and higher-sugar snacks. In terms of food intake, the hummus snack led to greater dietary compensation (of the snack) compared to the higher-sugar snack (*p* = 0.05), and also reduced evening snacking calories compared to no snack (*p* < 0.05). However, this did not translate into reductions in total (daily) food intake.

With respect to chickpeas, two different repeated-measures studies in healthy young women found that 200 g (~220 kcal) or 556 g (~600 kcal) of chickpea consumption led to immediate (acute) reductions in postprandial appetite (at 45 and 60 min; both, *p* < 0.05) compared to white bread consumption matched for available carbohydrates [21,71]. However, these reductions were not sustained over the 2-h testing period. Similarly, the previously discussed Wong et al. study [22] found no difference in two-hour postprandial appetite following 300 kcal consumption of chickpeas or white bread. To date, only one previously mentioned study exists examining the longer-term impact of chickpea consumption on appetite control and satiety. The incorporation of an average of 104 g/d of chickpeas into the diet for 12 weeks resulted in significantly higher satiation ratings (*p* < 0.001) compared to when they consumed their habitual diet [13].

A number of acute studies exist examining subsequent meal food intake. In both of the Zafar et al. studies [21,71], chickpea consumption in one meal reduced subsequent food intake by ~200 kcals compared to both white bread (both, *p* < 0.001) [21,71] and mashed potatoes (*p* < 0.0001) [71]. However, the Wong et al. study [22] found no effect on subsequent energy intake when comparing chickpea to white bread consumption. Lastly, the 12-week longer-term study by Murty et al. [13] reported no differences in daily intake when comparing chickpeas vs. habitual diet.

Collectively, these data illustrate mixed findings with some supporting improvements in appetite control, satiety, and food intake following the acute consumption of hummus and/or chickpeas while others report no effects. Given that only one long-term study exists, further work is needed to assess the impact of hummus on weight management.

## 7. Hummus within Healthy Dietary Patterns

Hummus originated in the Middle East and Mediterranean regions. The earliest recipe of chickpea-based hummus with tahini is dated back to the thirteenth-century [72]. The traditional dietary patterns of these regions focused on consuming a variety of plant-based foods such as fruits, minimally processed whole grains, vegetables, legumes, nuts, and seeds [73]. Thus, hummus was and continues to be a staple within the Middle East and Mediterranean dietary patterns. For example, in Israel, nearly 70% of the population is thought to have hummus in their refrigerator, and over 90% of Israelis will eat hummus throughout the week [74]. Those that continue to closely adhere to traditional Middle Eastern or Mediterranean diets have a lower risk than the general population of developing obesity, atherosclerosis, CVD, cancer, diabetes, and respiratory, kidney, or neurodegenerative diseases [73,75,76]. Further, a study conducted in Israel found that high consumers of legumes (i.e., those more likely to be eating greater amounts of hummus and better adhering to a Mediterranean diet) are at a reduced risk of CVD (*p* = 0.014) compared to the low consumers of legumes [77]. Several other studies have found inverse associations between legume intake and CVD, colorectal adenoma, and mortality [78,79,80,81].

In the 2015–2020 Dietary Guidelines for Americans, the Mediterranean-Style eating pattern was recommended as one example of a dietary pattern that is associated with positive health outcomes [4]. In a meta-analysis conducted on six Mediterranean diet studies in the United States, a 22% reduction (RR = 0.78, 95% CI 0.76–0.80, *p* = 0.02) in mortality rates was found among the highest adherers to the Mediterranean diet as compared to the lowest adherers [82]. Meta-analyses conducted on studies from around the world (including many studies from the United States) demonstrate that high Mediterranean diet adherence significantly reduces risk of diabetes (RR = 0.81, 95% CI 0.73–0.90, *p* < 0.0001) compared to low adherence [83] and CVD (RR = 0.81, 95% CI 0.74–0.88, *p* < 0.05) [84]. Additionally, compared with a control diet, consumption of a Mediterranean diet led to greater weight loss (mean difference, −1.75 kg, 95% CI −2.86 to −0.64 kg, *p* < 0.001), especially when this diet is combined with energy restriction or increased physical activity [85]. Overall, a Mediterranean-style diet appears to be beneficial to promote health and well-being. 

Since hummus and hummus ingredients contribute to improvements in health outcomes and have been included as part of a Mediterranean-style diet outside of the United States, it is important to develop food patterning and menu modeling to translate the research and recommendations into dietary strategies that can be included as part of a healthy American diet.

## 8. Translating the Evidence into Practice—Incorporating Legumes and Hummus in Healthy Dietary Patterns

In addition to the Healthy Mediterranean-style eating pattern, the 2015–2020 Dietary Guidelines for Americans also recommend the Healthy U.S. Style and Healthy Vegetarian eating patterns, and each pattern includes recommended weekly servings of legumes (beans and peas). To quantify, these recommendations include 1.5 cups of beans per week (for a 2000 calorie diet) as part of an overall healthy diet [4]. Incorporating legumes can be easy, with small and simple substitutions or additions to favorite meals and snacks. For example, chickpeas can be seasoned and served as a side dish, added to casseroles or soups or sprinkled on salads. Similarly, hummus can be easily served as part of meals like hummus bowls, hummus flatbreads, and hummus toasts, offering variety and contributing legume servings from the chickpea base. Hummus can also serve as a substitute for foods higher in saturated fat, sodium, or added sugars such as substituting mayonnaise on sandwiches. In general, 2 to 3 ounces of hummus is equivalent to about 1/8–1/4 cup of legumes.

Food patterning is one tool to demonstrate how to incorporate legumes and hummus into daily diets. An excellent example is USDA’s MyPlate 2-week sample menus for families wanting to eat healthy on a budget. The menus include legumes and hummus to meet recommended weekly legume servings and nutritional targets [86].

Menu modeling is another tool to demonstrate how simple dietary additions or substitutions can have a strong nutritional impact. For example, substituting chickpeas and/or hummus for other common ingredients can improve the nutrient density of the diet by increasing some beneficial nutrients like fiber and decreasing calories and other nutrients that are important to keep in check, like saturated fat, sodium, and added sugars. This is illustrated in the one-day menu in Table 2. Simply substituting hummus for some energy-dense but nutrient-poor foods and adding 1/3 cup of chickpeas contributes almost 1/2 cup of legumes, increases fiber and protein by 7 g, and also increases calcium, iron, and potassium in the daily diet. Other positive dietary changes included 90 less calories, 4 g less saturated fat, 35 mg less sodium, and 9 g less added sugars in the day. As illustrated by these examples, dietary change does not need to be extreme to deliver positive nutritional impact.

## 9. Conclusions

In the last few decades, hummus has become increasingly popular in the United States due to its versatility and perceived health benefits. This comprehensive review examined the effects of hummus, and its associated ingredients on risk factors related to type 2 diabetes, cardiovascular disease, and obesity. Overall, the consumption of hummus is associated with improved nutrient intake, diet quality, and healthier eating habits. In addition, hummus consumption is demonstrated to improve glucose control over the short-term compared to other commonly consumed foods such as white bread. These benefits may be a result of the nutrients within hummus, including protein, fiber, resistant starch, unsaturated fats, and numerous polyphenols. Additionally, incorporation of chickpeas or tahini into the diet is shown to improve long-term glycemic response and promote cardiovascular health through the lowering of cholesterol, lipid, and blood pressure levels. While future studies are needed to investigate the exact role of hummus consumption on health and disease, collectively, the current evidence supports the consumption of hummus, or hummus ingredients, as part of a healthy diet to improve health.

## Figures and Tables

**Table 1 nutrients-12-03678-t001:** Nutrient profile for chickpeas, tahini, and hummus.

Nutrient	Value per Serving and per 100 g ^a^						
	Chickpeas, Cooked, Boiled without Salt		Sesame Butter, Tahini, from Roasted/Toasted Kernels		Hummus, Commercial		
FoodData Central ID	173757		170189		174289		
Serving Size	1 Cup (164 g)	100 g	2 tbsp (30 g)	100 g	2 tbsp (28 g)	% Daily Value	100 g
Macronutrients							
Energy, kcal	269	164	179	595	67		237
Protein, g	14.5	8.9	5.1	17	2.21	~3%	7.78
Fat, g	4.25	2.6	16.1	53.8	5.05	6%	17.82
Carbohydrate, g	45	27.4	6.4	21.19	4.25	2%	15
Fiber, g	12.5	7.6	2.8	9.3	1.56	6%	5.5
Sugar, g	7.87	4.8	0.15	0.49	0.18		0.62
Added Sugar, g	0	0	0	0	0	0%	0
Minerals							
Calcium, mg	80	49	128	426	13	1%	47
Iron, mg	4.74	2.89	2.69	8.95	0.72	4%	2.54
Magnesium, mg	78	48	29	95	21	5%	75
Phosphorus, mg	276	168	220	732	51	4%	181
Potassium, mg	477	291	124	414	88	2%	312
Sodium, mg	11.5	7	35	115	121	5%	426
Zinc, mg	2.5	1.53	1.39	4.62	0.4	4%	1.44
Copper, mg	0.58	0.35	0.48	1.61	0.11	12%	0.38
Manganese, mg	1.69	1	0.44	1.45	0.33	14%	1.16
Selenium, μg	6.07	3.7	10.32	34.4	1.33	2%	4.7
Vitamins							
Vitamin C, mg	2.13	1.3	0	0	0	0%	0
Thiamin, mg	0.19	0.12	0.36	1.22	0.05	4%	0.16
Riboflavin, mg	0.1	0.06	0.14	0.47	0.04	3%	0.13
Niacin, mg NE	0.86	0.52	1.63	5.45	0.29	2%	1
Pantothenic acid, mg	0.47	0.29	0.2	0.69	0.1	2%	0.35
Vitamin B6, mg	0.23	0.14	0.04	0.15	0.04	2%	0.15
Folate, mg DFE	282	172	29	98	13	3%	48
Choline, mg	70.2	42.8	7.7	25.8	NR ^(b)^		NR ^(b)^
Vitamin B12, μg	0	0	0	0	0	0%	0
Vitamin A, μg RAE	1.6	1	0.9	3	0.28	0%	1
Vitamin D, μg	0	0	0	0	0	0%	0
Vitamin K, μg	6.56	4	0	0	6.46	5%	22.8
Vitamin E, mg	0.57	0.35	0.07	0.25	0.44	3%	1.54
Lipids							
Saturated, g	0.44	0.27	2.26	7.53	0.73	4%	2.56
Monosaturated, g	0.96	0.58	6	20.3	1.5		5.34
Polyunsaturated, g	1.9	1.2	7	23.56	2.5		8.8

^a^ Data obtained from the USDA FoodData Central; ^b^ NR = not reported.

**Table 2 nutrients-12-03678-t002:** One-day menu model example of hummus inclusion into the American diet.

	Standard Meal	Revised Meal	Overall Change in Daily Nutrients
Breakfast	Toast with 1 tsp butter and 1 Tbsp jam	Toast with 1 tbsp hummus	↑ ~½ cup legumes ↑ 7 g protein ↑ 7 g fiber ↑ calcium, iron, potassium ↓ 90 kcal ↓ 4 g saturated fat ↓ 35 mg sodium ↓ 9 g added sugar
Lunch	Sandwich with 2 tbsp mayonnaise	Sandwich with 2 tbsp hummus	
Dinner	Salad with salad dressing	Salad with salad dressing and topped with 1/3 cup chickpeas	

↑ indicates amount increase in daily nutrients; ↓ indicates amount decrease in daily nutrients.

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
