# Peer review of "The Benefits of Including Hummus and Hummus Ingredients into the American Diet to Promote Diet Quality and Health: A Comprehensive Review"

_nutrients, 2020, doi:10.3390/nu12123678_

Round 1
Reviewer 1 Report
In their comprehensive review Reister et al. expose the effect of hummus and hummus ingredient consumption as a means to promote diet quality and health, with particular focus on prevention of type 2 diabetes, obesity and cardiovascular disease. They also suggest a role for hummus in dietary guidelines.
In the Introduction section, reasons for the increasing presence of hummus in the American diet are considered, along with a possible use of hummus consumption in place of vegetables, which consume remains low in the American population.
In the next paragraph, hummus ingredients and their nutrient support and favorable effects are explained. Dedicated studies are also reported.
In the following sections, the effects of hummus in the diet to control glycemia, lipids, and obesity, and related studies are properly illustrated. In the last part of the review, hummus as a component of healthy dietary pattern and its use suggestion are treated.
Tables are adequate.
I found the review very interesting and pleasant to read. I believe that the title satisfactorily anticipates the paragraphs debated in the paper.
Therefore, I have no suggestions to improve this manuscript, that in my opinion encounters the expectations of the reader.
Page 3, line 73: particular in capital letter.
Author Response
We appreciate the time spent in reviewing this manuscript and providing comments and suggestions. We have highlighted all the text in the manuscript that has been revised. Our specific responses to each comment by the reviewers are included below.
Reviewer 1: Summary Comment:
I found the review very interesting and pleasant to read. I believe that the title satisfactorily anticipates the paragraphs debated in the paper. Therefore, I have no suggestions to improve this manuscript, that in my opinion encounters the expectations of the reader.
Authors’ Response:
We appreciate your kind and thoughtful comments.
Reviewer 1 Minor Comment:
Page 3, line 73: particular in capital letter.
Authors’ Response:
This has been corrected.

Reviewer 2 Report
The idea of considering chickpea and tahini ingrédients is of interest egarding their nutrient contents. Actually, the paper in centerd on the so-called hummus.
Although the method used is correct, the hypothesis goes in a way against the basic principles in nutrition which consist in considering food groups in an overall balance, this approach is quite missing. So you could try to consider hummus as belonging to a wider group of food and simulate the global regimen and nutrient intakes taking it into account - but not a contrario, introduce hummus in a first intention as hummus is probably substitutable by mutiple complementar foods.
You have to question the introduction in American FBDG as the multicultural society could impact the acceptabilty of this product in its diet.
References regarding methods to establish FBDG are missing, and you could maybe discuss deeper the methods to establish FBDG and how to introduce a specific type of product or ingredients in balance with the others.
As a detail: it would have been interesting to complete the composition table by the percentage of daily requirements
Author Response
We appreciate the time spent in reviewing this manuscript and providing comments and suggestions. We have highlighted all the text in the manuscript that has been revised. Our specific responses to each comment by the reviewers are included below.
Reviewer #2 Comment:
The idea of considering chickpea and tahini ingrédients is of interest egarding their nutrient contents. Actually, the paper in centerd on the so-called hummus.
Authors’ Response:
The paper examines the evidence concerning hummus (and the respective chickpea and tahini ingredients) on diet quality and health. To clarify, hummus is not a novel food. It is a documented food categorized within the UDSA Food Data Central database.
Reviewer #2 Comment:
Although the method used is correct, the hypothesis goes in a way against the basic principles in nutrition which consist in considering food groups in an overall balance, this approach is quite missing.
Authors’ Response:
The Dietary Guidelines for Americans serves as the cornerstone of federal nutrition programs and policies, providing food-based recommendations through scientific evidence to help prevent diet-related chronic diseases and promote overall health. While the current guidelines highlight several healthy dietary patterns, like the Mediterranean Diet, the recommendations focus on specific foods and/or food groups. For example, vegetables, fruits, and dairy – which are food groups are all part of the recommendations. However, added sugars, sugar-sweetened beverages, red meat, etc. are not food groups but actual foods that are also examined given their health implications. This review follows a similar pattern in that we examine whether the consumption of hummus (or hummus ingredients) improve health outcomes. We then highlight how hummus has already been part of the Mediterranean style dietary pattern and how it might be included into the American diet. Section #6 specifically highlights the healthy dietary pattern first and then summarizes the inclusion of hummus into this pattern.
Reviewer #2 Comment:
So you could try to consider hummus as belonging to a wider group of food and simulate the global regimen and nutrient intakes taking it into account - but not a contrario, introduce hummus in a first intention as hummus is probably substitutable by mutiple complementar foods.
Authors’ Response:
We apologize for our inability to understand this point due to some typos and/or confusing phrases.
We will comment that we do consider hummus as belonging to a wider group of foods. As stated on lines 36-29 in Section #1:
Further, hummus fits into a variety of healthy eating patterns, including the Mediterranean dietary pattern [3], and is used as part of the protein foods and vegetable categories within USDA’s MyPlate [4].
In addition, on lines 40-51 in Section #1, we state:
Americans continue to consume less vegetables than the current dietary guideline recommendations. In 2015, only one in 10 Americans met the 2015-2020 Dietary Guidelines for Americans’ vegetable recommendations of 2.5 cups/day within a 2,000 calorie diet [5][4]. Within these guidelines, it is recommended that at least 1.5 cups/wk of vegetables should be from legumes; however, mean intake of legumes is less than half of the recommended amount in American adults [4]. Although one serving (i.e., two tablespoons) of hummus per day could contribute to meeting the weekly recommendations, it is unclear whether there is sufficient evidence supporting the specific effects of hummus on health outcomes.
Lastly, on lines 297-301 in Section #6, we state:
Since hummus and hummus ingredients contribute to improvements in health outcomes and have been included as part of a Mediterranean-style diet outside of the United States, it is important to develop food patterning and menu modeling to translate the research and recommendations into dietary strategies that can be included as part of a healthy American diet.
Thus, we feel that we have adequately discussed hummus as part of key vegetable and protein foods included within the healthy dietary patterns, like the Mediterranean diet, recommended within he Dietary Guidelines.
Reviewer #2 Comment:
You have to question the introduction in American FBDG as the multicultural society could impact the acceptability of this product in its diet.
Authors’ Response:
Hummus was introduced into the American diet in the early 1940s. However, it has grown increasingly more common over the past 5 years. Current estimates suggest that 25% of American households have hummus in their refrigerators. This surge in hummus purchasing might be a result of the 2015 Dietary Guidelines highlighting the Mediterranean Diet, which includes hummus, as one of the dietary patterns recommended for improved health. This review was designed to summarize the evidence supporting hummus’ role to promote health.
Reviewer #2 Comment:
References regarding methods to establish FBDG are missing, and you could maybe discuss deeper the methods to establish FBDG and how to introduce a specific type of product or ingredients in balance with the others.
Authors’ Response:
The purpose of this review is not to establish food-based dietary guidelines. It is designed to critically evaluate the effects of consuming hummus on health outcomes related to obesity, type 2 diabetes, and cardiovascular risk factors. While we appreciate this comment, it is out of scope of this review. With that said, we do include a food patterning modeling example to illustrate the inclusion of hummus into the diet. As stated on lines 304 – 315 of section #7:
In addition to the Healthy Mediterranean-style eating pattern, the 2015-2020 Dietary Guidelines for Americans also recommend the Healthy U.S. Style and Healthy Vegetarian eating patterns, and each pattern includes recommended weekly servings of legumes (beans and peas). To quantify, these recommendations include 1.5 cups of beans per week (for a 2000 calorie diet) as part of an overall healthy diet. Incorporating legumes can be easy, with small and simple substitutions or additions to favorite meals and snacks. For example, chickpeas can be seasoned and served as a side dish, added to casseroles or soups or sprinkled on salads. Similarly, hummus can be easily served as part of meals like hummus bowls, hummus flatbreads and hummus toasts, offering variety and contributing legume servings from the chickpea base. Hummus can also serve as a substitute for foods higher in saturated fat, sodium or added sugars such as substituting mayonnaise on sandwiches. In general, 2 to 3 ounces of hummus is equivalent to about 1/8 – 1/4 cup of legumes.
Reviewer #2 Comment:
As a detail: it would have been interesting to complete the composition table by the percentage of daily requirements
Authors’ Response:
We have added the percentage of daily requirements to Table 1. Nutrient profile for chickpeas, tahini, and hummus
